# TiEBe: Tracking Language Model Recall of Notable Worldwide Events Through Time

## Abstract

As the knowledge landscape evolves and large language models (LLMs) become increasingly widespread, there is a growing need to keep these models updated with current events. While existing benchmarks assess general factual recall, few studies explore how LLMs retain knowledge over time or across different regions. To address these gaps, we present the Timely Events Benchmark (TiEBe)—a dataset of over 23,000 question–answer pairs centered on notable global and regional events, spanning more than 10 years of events, 23 regions, and 13 languages. TiEBe leverages structured retrospective data from Wikipedia to identify notable events through time. These events are then used to construct a benchmark to evaluate LLMs' understanding of global and regional developments, grounded in factual evidence beyond Wikipedia itself. Our results reveal significant geographic disparities in factual recall, emphasizing the need for more balanced global representation in LLM training. We also observe a Pearson correlation of more than 0.7 between models' performance in TiEBe and various countries' socioeconomic indicators, such as HDI. In addition, we examine the impact of language on factual recall by posing questions in the native language of the region where each event occurred, uncovering substantial performance gaps for low-resource languages.

## 1 Introduction

Large language models (LLMs) have rapidly become central to numerous applications Eloundou et al. (2023); Noy & Zhang (2023); Hadi et al. (2023), prompting continuous efforts to refine and update them. Keeping these models' knowledge timely and accurate as the world's events unfold has grown increasingly important. Continual pretraining Zhang et al. (2023); Wu et al. (2024); Gogoulou et al. (2024) has emerged as a promising paradigm for systematically integrating new information, ensuring that models remain current with ongoing global affairs. However, despite clear interest in dynamically updating LLMs, there remains a shortage of a dedicated and continuously evolving benchmark to measure how well these models capture and retain factual knowledge of major world events over time.

Another critical challenge in evaluating LLMs lies in the significant regional disparities in their performance Sathish et al. (2024); Kantharuban et al. (2023). Research has shown that LLMs often exhibit stronger factual recall for content originating in certain regions, typically those well-represented in their training datasets, while underperforming on data from less-represented areas Moayeri et al. (2024); Myung et al. (2024). Despite these known disparities, the number of benchmarks designed explicitly to assess and quantify these regional gaps is limited. This lack of evaluation tools hinders our ability to understand and address the inequalities in how LLMs process and recall information about different parts of the world.

To address these challenges, we introduce the Timely Events Benchmark (TiEBe), a benchmark designed to evaluate an LLM's knowledge of noteworthy events worldwide and at the regional level. Our approach leverages structured information from Wikipedia retrospective pages to identify external data sources, which we then use to generate question-answer (QA) pairs that reflect notorious occurrences in a given year and a given region. This strategy enables us to continuously assess a model's knowledge of evolving global affairs while also measuring geographical disparities. Furthermore, by relying on publicly available Wikipedia data that is naturally updated, TiEBe can be

easily and regularly updated, ensuring that evaluations remain aligned with current world events and that models can be reassessed as new events unfold. Our results demonstrate substantial regional disparities in factual recall across all LLMs tested, highlighting the critical need for improvements in this area.

The main contributions of our paper are as follows:

- We introduce TiEBe, a benchmark of more than 23 thousand question-answer pairs grounded on noteworthy events, spanning 10 years, 13 languages and 23 different geographic regions.
- TiEBe provides the QA pairs for non-English speaking countries in both English and in their native languages.
- We perform various evaluations to measure LLM factual recall over time, different regions, and languages, and find some notable performance gaps.

## 2 RELATED WORK

As large language models (LLMs) continue to improve, there is growing interest in evaluating their ability to comprehend and recall factual knowledge about the world. Although many studies have investigated LLMs' capacity for general factual recall Mallen et al. (2022); Tang et al. (2022); Wei et al. (2024), it has become evident that this ability varies significantly based on the geographic or cultural context of the data. For example, WorldBench Moayeri et al. (2024) highlights regional disparities in LLM performance, demonstrating that their ability to recall facts about local economic and social statistics can differ significantly depending on the region. BLEND Myung et al. (2024) demonstrates a notable difference in LLM performance when prompted about cultural aspects of different countries, both in English and in the native languages of the countries. In addition, Multi-FAct Shafayat et al. (2024) examines multilingual factuality, highlighting that the factual accuracy of LLMs differs between languages and exhibits a bias towards western-centric information. Our work expands on these types of evaluations by focusing on notorious events—historical and significant occurrences—associated with specific countries or with global impact. By emphasizing in events, our dataset uniquely evaluates factual knowledge through time and regions.

In parallel, the paradigm of continual learning has gained traction as a cost-effective alternative to retraining models from scratch Wu et al. (2024). This approach seeks to enable LLMs to incorporate new knowledge without forgetting previously learned information, a challenge known as catastrophic forgetting Ibrahim et al. (2024); Zhai et al. (2023). Despite its promise, the field still suffers from a limited number of diverse benchmarks for evaluating how well models balance learning new content with retaining existing knowledge White et al. (2024); Jain et al. (2024). To address this, TemporalWiki Jang et al. (2022) proposes a benchmark based on tracking changes in Wikipedia articles, allowing researchers to assess how well LLMs adapt to evolving world knowledge. While TemporalWiki focuses on factual updates in encyclopedic content, our work complements it by evaluating LLMs' understanding of events across time and geography.

## 3 METHODOLOGY

In this section, we describe the pipeline for creating TiEBe, which is illustrated in Figure 1.

### 3.1 DATA COLLECTION

A Wikipedia retrospective is a page that lists and summarizes notable events from a specific year in a given country, domain, or globally. Each event also typically cites a few external sources, usually new articles, providing further context. We leveraged such pages by extracting events and their corresponding sources.

To study factual recall over time, we used a 10-year timespan, covering retrospective pages from 2015 to April 2025. We selected retrospective pages from 23 regions: 22 countries and one global category ("World") that includes events of broad international relevance. The countries are grouped as follows:

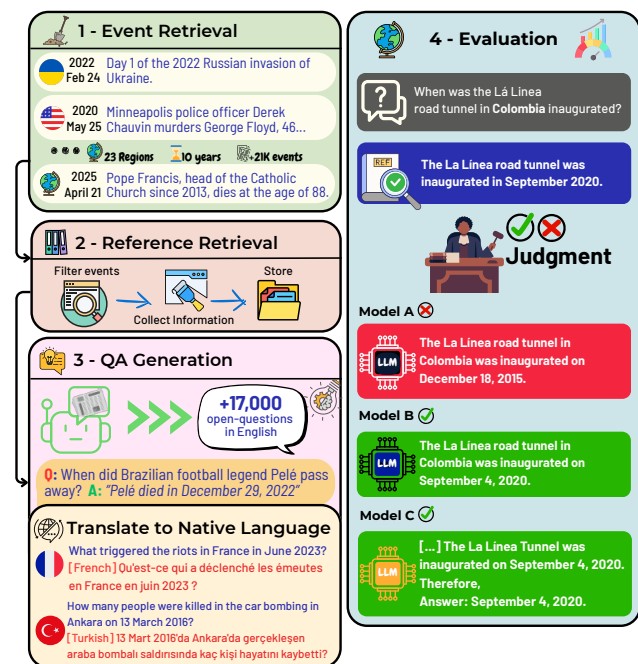

Figure 1: Illustration of the pipeline used to build TiEBe.

- **North America** – United States, Canada, Mexico
- **South America** – Brazil, Argentina, Colombia
- **Asia** – India, China, Indonesia
- **Oceania** – Australia, Papua New Guinea, New Zealand
- **Western Europe** – Germany, United Kingdom, France, Portugal
- **Eastern Europe** – Russia, Ukraine, Turkey
- **Africa** – Nigeria, Democratic Republic of the Congo, Ethiopia

We included the three most populous countries from each macro-region, except for Portugal, which was added because it shares a language with Brazil, and we are evaluating models specialized in Portuguese. Together, the selected countries represent over half of the world's population.

We try to retrieve as many sources cited in the events as possible; however, many cited sources are no longer available or do not allow scraping of their contents. Overall, we collected 21k events from all retrospective pages, but we were able to gather external references for only 17370 events. More details about the collection process can be found in the Appendix C.2.

### 3.2 GENERATION OF QUESTION-ANSWER PAIRS

From each source document cited in the event descriptions, we generate synthetic question–answer (QA) pairs focused on the events discussed. These QA pairs are designed to test whether a model can recall the factual information present in the original source document.

Initially, all QA pairs were generated in English, even when the underlying documents were written in other languages. This choice enables us to isolate and evaluate factual recall without confounding effects from multilingual understanding. To generate the questions, we used DeepSeek-V3 Liu et al. (2024), providing it with the event description, the corresponding source document, and the date of the event. The complete prompt used for this generation process is included in Appendix B.1.

Figure 2 presents examples of the generated QA pairs across different regions, illustrating the wide range of topics covered.

Table 1: TiEBe Question distribution. The totals include questions in both English and the country's native language (for non-English speaking regions)

| Region | Language | 2015-2017 | 2018-2020 | 2021-2022 | 2023-2025 | Questions |
|--------|----------|-----------|-----------|-----------|-----------|-----------|
| Argentina | Spanish | 18 | 84 | 20 | 148 | 270 |
| Australia | English | 99 | 105 | 74 | 533 | 811 |
| Brazil | Portuguese | 254 | 204 | 278 | 338 | 1074 |
| Canada | English | 37 | 49 | 70 | 159 | 315 |
| China | Chinese | 56 | 60 | 204 | 294 | 614 |
| Colombia | Spanish | 8 | 58 | 26 | 138 | 230 |
| DR Congo | French | 22 | 6 | 34 | 226 | 288 |
| Ethiopia | Amharic | 36 | 86 | 68 | 150 | 340 |
| France | French | 54 | 36 | 24 | 398 | 512 |
| Germany | German | 16 | 48 | 22 | 360 | 446 |
| India | Hindi | 152 | 102 | 292 | 554 | 1100 |
| Indonesia | Indonesian | 490 | 1052 | 660 | 476 | 2678 |
| Mexico | Spanish | 50 | 1460 | 150 | 304 | 1964 |
| New Zealand | English | 24 | 77 | 98 | 650 | 849 |
| Nigeria | English | 22 | 64 | 44 | 118 | 248 |
| Papua New Guinea | Tok Pisin | 12 | 6 | 50 | 50 | 118 |
| Portugal | Portuguese | 110 | 252 | 28 | 66 | 456 |
| Russia | Russian | 62 | 62 | 56 | 544 | 724 |
| Turkey | Turkish | 62 | 220 | 190 | 286 | 758 |
| Ukraine | Ukrainian | 30 | 16 | 208 | 326 | 580 |
| United Kingdom | English | 487 | 873 | 880 | 2002 | 4242 |
| United States | English | 522 | 900 | 800 | 1006 | 3228 |
| World | — | 164 | 639 | 345 | 453 | 1601 |
| **Total** | **—** | **2787** | **6459** | **4621** | **9579** | **23446** |

To assess the impact of language on model performance, we also translated the questions into the native languages of non-English-speaking countries. This allows us to analyze how well models perform under language shift. These translations were also produced using DeepSeek-V3, which showed stronger performance in our preliminary evaluations. Table 1 shows the question distribution per country and year. In total, we arrive at 23446 QA pairs, 17370 in English, and 6076 in the native languages of the respective countries.

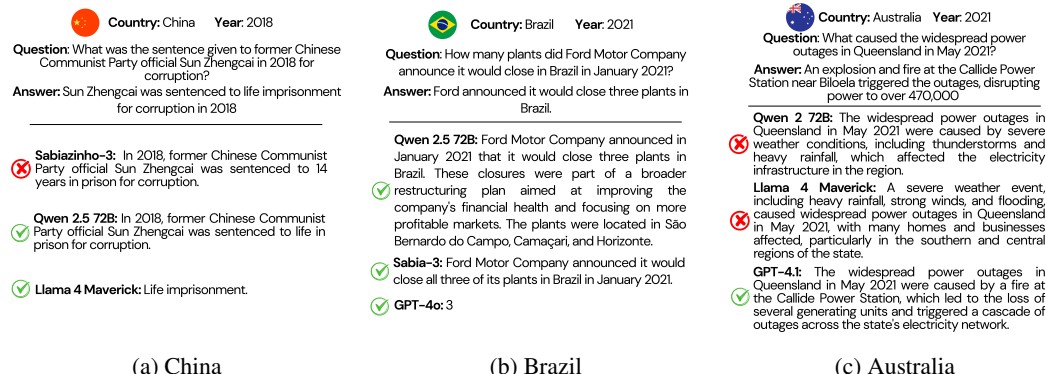

(a) China   (b) Brazil   (c) Australia

Figure 2: Examples of generated question-answer pairs by country.

## 3.3 MODEL EVALUATION

We evaluated nine different models: three open-source—Qwen 2 72B, Qwen 2.5 72B, and Llama 4 Maverick—and six commercial models. The commercial models include Sabiá-3 and Sabiazinho-3 Abonizio et al. (2024) from Maritaca AI, Mistral-large from Mistral, and GPT-4o, GPT-4.1-mini, and GPT-4.1 OpenAI (2024) from OpenAI. Several of these models have a regional or linguistic

Table 2: Comparison of Model-as-Judge vs. Human Judgment on 200 Samples.

|  | % Judged As Correct | % of divergences with human |
|---|---|---|
| Human | 58.5% | - |
| Deepseek | 53.0% | 11.5% |
| GPT-4o | 54.5% | 9.0% |

focus. For instance, the Qwen models prioritize Chinese data, Sabiá-3 is primarily trained on Brazilian data, and Mistral-large highlights strong performance in European languages such as French and German. Llama 4 and the OpenAI models serve as strong baselines representing the current state of the art in open-source and proprietary systems.

All models are evaluated in the same manner. Each question is provided to the LLM as a zero-shot prompt. We then use an LLM-as-judge Gu et al. (2024); Zheng et al. (2023); Li et al. (2024) to evaluate the answer of each model. In this study, we use DeepSeek-V3 as the judge. The judge receives the question, the candidate's answer provided by the LLM, and the expected answer created previously in our QA generation process. The judge then decides whether the provided answer is correct or not. The full prompt used for the judge can be found in Appendix B.1.

All model inferences were performed using APIs. For more detailed information about the tested models, please refer to Appendix B.2.

### 3.4 LLM-AS-JUDGE PERFORMANCE

To assess the reliability of Deepseek-v3 as an automatic judge of model responses, we manually annotated 200 randomly sampled questions from TiEBe. We randomly selected a single answer from one of the candidate models for each question.

Table 2 presents the agreement rates between human annotations and those made by Deepseek-v3 and GPT-4o. Deepseek-v3 matched human judgment in 88.5% of the cases, while GPT-4o achieved a slightly higher agreement rate of 91%. In general, both models tended to be stricter than the human annotator, often marking as incorrect answers accepted by the human.

## 4 RESULTS

This section will discuss the results of the 9 tested models in the TiEBe dataset, exploring overall accuracy and their regional and temporal performance.

### 4.1 REGIONAL PERFORMANCE

Figure 3 presents the accuracy of each tested model across all regions, under two conditions: (a) considering only events that occurred before 2023, and (b) using the full set of events. Focusing on pre-2023 events is particularly informative, as all models have a training cutoff after that date, ensuring a fairer basis for comparison.

**Large regional performance disparities exist across all models**. Among the 22 countries tested in TiEBe, 12 show a performance gap of at least 20 percentage points compared to the United States. The largest observed gap is 41 points, notably in regions such as the Democratic Republic of Congo. Even when focusing only on events that occurred before 2023—thus excluding potential advantages from more recent training data—9 countries still show gaps of 20 points or more, with the maximum reaching 40 points. These findings highlight a consistent imbalance in factual recall across geographic regions in all tested models.

**Model performance is positively correlated with country GDP.** When evaluating only the events that took place before all models' training cutoff dates, we find a strong correlation between a country's GDP and model performance. Specifically, the average performance across models correlates

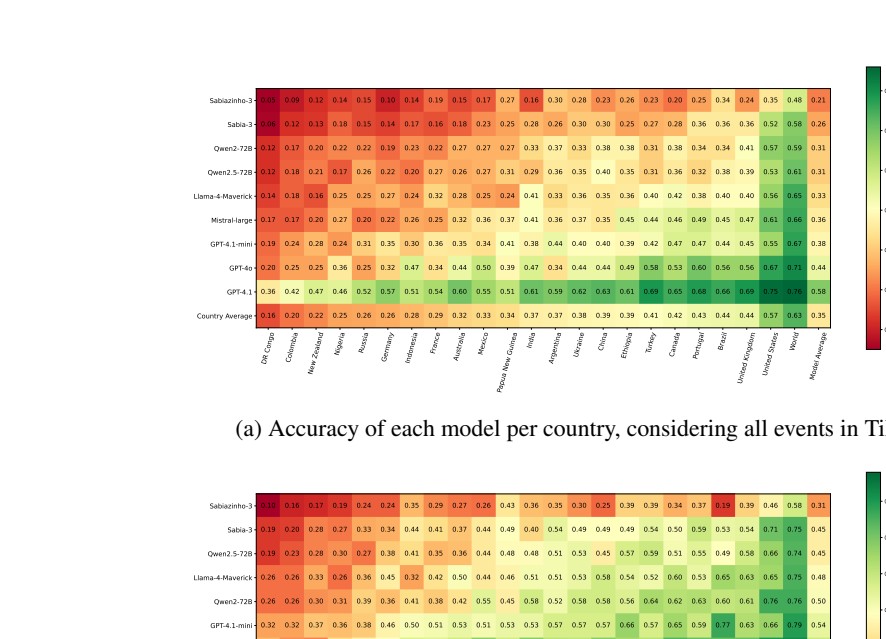

(a) Accuracy of each model per country, considering all events in TiEBe.

(b) Accuracy of each model per country, considering only events before 2023.

Figure 3: Performance of models per country under different subsets of TiEBe.

with GDP at a Spearman coefficient of 0.73. This suggests that models tend to recall information more accurately for wealthier countries, indicating possible socioeconomic bias in training data.

**GPT-4.1 achieves the highest performance among all models tested.** On the full dataset, GPT-4.1 significantly outperforms all other models, with a 14-point lead over the second-best model, GPT-4o. This advantage is largely due to its more recent training cutoff, as the gap between the two models drops to just 2 points when considering only pre-2023 events. This implies that GPT-4.1 incorporates more recent knowledge but does not significantly improve earlier events. Overall, GPT-4.1 outperforms the best non-OpenAI model (Mistral-large) by at least 15 percentage points in average accuracy. However, despite the good performance, GPT-4.1 still shows significant regional gaps in factual recall.

## 4.2 TEMPORAL PERFORMANCE

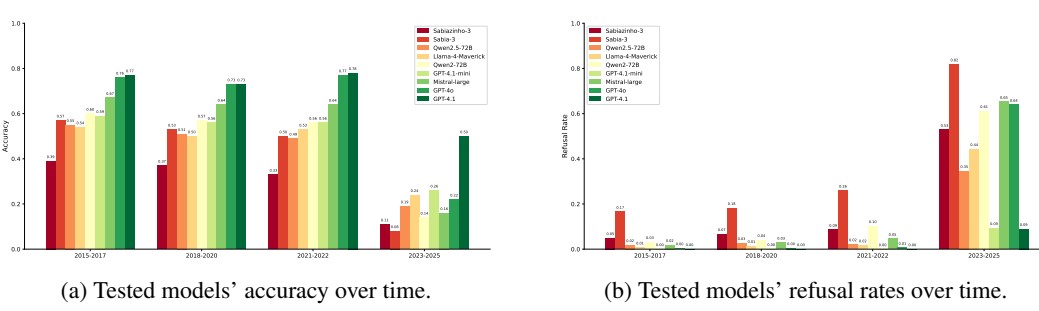

(a) Tested models' accuracy over time.

(b) Tested models' refusal rates over time.

Figure 4: Accuracy and refusal rates over different time periods.

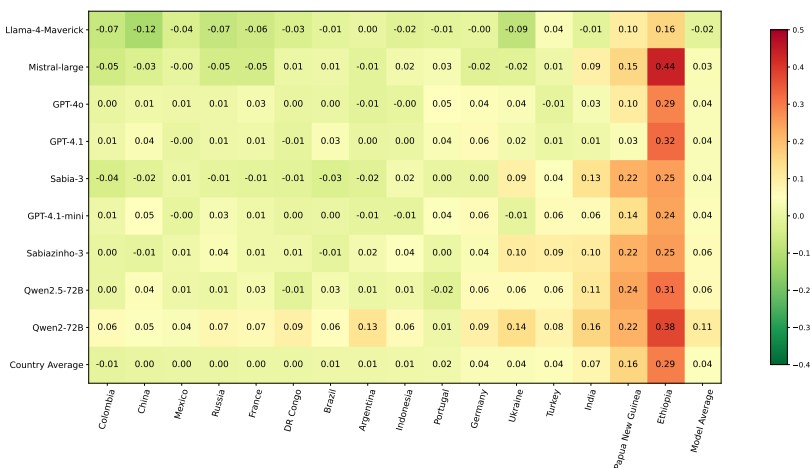

Figure 5: Difference in overall accuracy when prompted in English or the country native language. Negative value means the accuracy of the model was lower in the native than in English, while positives value indicate that models performed better in the native language.

We also examine how model performance varies across different time periods. Figure 4 shows the accuracy and refusal rates of all models across four intervals: 2015–2017, 2018–2020, 2021–2022, and 2023–2025. Detailed yearly results for each model are provided in the Appendix D.

Among the models, GPT-4o and Mistral-large report a knowledge cutoff in October 2023. Qwen 2 72B, Qwen 2.5 72B, Sabiá-3, and Sabiazinho-3 list 2023 as their cutoff year without specifying a month. LLaMA 4 Maverick reports a cutoff in August 2024, while GPT-4.1 and GPT-4.1 mini are current up to July 2024.

Across the first three time periods (2015–2022), model performance remains relatively stable. However, there is a notable drop in accuracy from 2023 to 2025, which aligns with the models' training cutoffs. During this final period, most models also exhibit a significant increase in refusal rates, as they refuse to answer questions beyond their training.

Notably, the Sabiá-3 model displays unusually high refusal rates, even for events that predate its reported cutoff. This behavior contributes to its lower overall performance.

### 4.3 THE EFFECTS OF MODEL LANGUAGE

With the translated questions for all non-English speaking regions, we repeated our experiment, regenerating all answers from each model and repeating all the judgments. We show in Figure 5 the difference in accuracy for each model and region, when comparing the accuracy with English questions with the accuracy with questions in the countries respective native language. Positive values indicate that the model performed better in the native language, while negative values indicate the model performed better in English.

**10 out of the 16 countries show an average performance difference of less than 3%.** This relatively small variation suggests that, for most regions, translating questions into the native language did not significantly affect model accuracy. In these cases, the models demonstrated comparable understanding of the content regardless of whether it was presented in English or the native language, indicating a degree of multilingual robustness.

**We notice big performance degradation for Tok Pisin and Amharic**, the languages of Papua New Guinea and Ethiopia, respectively. These two cases stand out as the most significant drops in accuracy across all evaluated models, adding to the body of evidence that current LLMs struggle to generalize well to very low-resource languages Magueresse et al. (2020); Joshi et al. (2020). The lack of sufficient Tok Pisin and Amharic representation in the models' training data likely contributes to this performance gap. In particular, even high-capacity commercial models, which generally maintain robust performance across other languages, failed to retain accuracy when an-

swering questions translated into these languages. This highlights a broader challenge in building equitable multilingual models that maintain performance in underrepresented linguistic contexts.

**Llama4 Maverick was the only model to show a slight performance increase in the native languages.** Unlike all other models, LLama4 Maverick achieved, on average, marginally better results when responding to questions in the respective native languages of the countries evaluated. This may reflect more effective multilingual pretraining or fine-tuning strategies, allowing the model to handle non-English inputs better.

**Qwen2-72B shows the biggest performance degradation on non-English questions.** Upon further investigation, we observed that this drop in accuracy is largely driven by an increased refusal rate when the model is prompted in non-English languages. Qwen2-72B often declines to respond altogether, significantly impacting its measured performance. This behavior suggests that the model may have a limited confidence threshold for non-English inputs or lacks sufficient multilingual alignment.

## 4.4 PERFORMANCE CORRELATION WITH SOCIOECONOMIC INDICATORS

Table 3: Pearson and Spearman correlations between the average performance of models before 2023 and numerous socioeconomic indicators. Indications marked with * were used on a log scale for Pearson calculations.

| | GDP* | HDI | MYS | Population* |
|---|---|---|---|---|
| Spearman | | | | |
| English | 0.728 | 0.549 | 0.526 | 0.146 |
| Native languages | 0.767 | 0.747 | 0.728 | 0.022 |
| Pearson | | | | |
| English | 0.562 | 0.518 | 0.448 | 0.192 |
| Native languages | 0.765 | 0.791 | 0.803 | 0.131 |

To further analyze the correlation between the average performance of the tested models in TiEBe, we used the subset of questions regarding events before 2023, eliminating the effects of model cutoff dates being reached. We considered two scenarios

- **English:** Where we considered the average performance of models when prompted in English.
- **Native languages:** Where we considered the average performance of models when prompted in the native language of each region, for example, we consider the performance models had in Amharic for Ethiopia.

Table 3 reports both Spearman and Pearson correlation coefficients for both scenarios between the average accuracy of all tested models with events before 2023 and four social and economic indicators, Gross Domestic Product (GDP), Human Development Index (HDI), Mean Years of Schooling (MYS), and Population. More detailed information about the collected statistics for each country and further analyses can be found in the appendix D.

**Model performance shows substantial correlation with economic and educational indicators.** The average accuracy of models in each country is notably correlated with GDP, HDI, and MYS, particularly when questions are presented in the native languages. The Spearman correlation between model accuracy and GDP reaches 0.77 in the native language setting, while HDI and MYS correlate at 0.75 and 0.73, respectively. These results suggest that LLMs tend to perform better in countries that are economically and educationally more developed, likely due to the higher availability and representation of such regions in the models' training data.

**Performance correlations are higher in native language evaluations.** Across all indicators, correlations are consistently stronger when models are evaluated using questions in the native language rather than in English. This indicates that regions with higher development levels receive more data coverage and more robust multilingual training data. In contrast, lower-resource countries may suffer a double penalty: underrepresentation and lack of training for their native languages.

**Population size shows weak correlation with performance in both scenarios.** Despite expectations that more populous countries might benefit from increased digital presence and, by extension, more training data, our analysis shows little correlation between population and model performance on TiEBe. Both Spearman and Pearson correlations remain low across English and native-language evaluations. This indicates that data representation in training corpora might be shaped more by economic and infrastructural factors than by sheer demographic size.

These findings show the presence of systemic imbalances in current LLMs, where performance is notably correlated to socioeconomic factors. Addressing such disparities will be essential for building more equitable and globally representative language technologies.

## 5 CONCLUSION

In this work, we introduced the Timely Events Benchmark (TiEBe), a large-scale evaluation framework designed to assess factual recall in LLMs across time, regions, and languages. TiEBe comprises over 23,000 question–answer pairs grounded in notable events extracted from Wikipedia retrospective pages, spanning a 10-year period and 23 geographic regions.

Our findings show that current LLMs exhibit considerable geographic disparities in factual recall. When considering only events before the cutoff date of all models, we observed a notable correlation of models' performance in TiEBe with socioeconomic indicators such as GDP, HDI, and MYS, suggesting that LLMs disproportionately favor wealthier, more digitally represented nations. Finally, we showed that while models show reasonably equal performance in most tested languages, performance still degrades sharply in low-resource languages, such as Tok Pisin and Amharic.

## 6 LIMITATIONS AND FUTURE WORK

Our dataset uses Wikipedia retrospective pages to identify and extract global and regional events. As a result, we cannot include regions where such pages are unavailable or contain too few documented events. Additionally, because all source documents are drawn from publicly accessible Wikipedia pages, there is a risk that some of the evaluated language models may have been exposed to this content during pretraining. While the overall results show substantial variation across models, potential contamination may influence performance and obscure true generalization capabilities.

Future work can address these limitations by incorporating events from broader sources, such as regional news archives. This would reduce contamination risks and improve the diversity of events and questions.

## 7 REPRODUCIBILITY STATEMENT

The code necessary to run the benchmark is available at `https://anonymous.4open.science/r/TiEBe-3235`. The methodology and components of TiEBe are explained in section 3. Further details that can be relevant for reproduction are discussed in Appendix B.

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

## A    USAGE OF LLMs

In the creation of this paper, LLMs were used to aid the writing process and to improve the flow of text. No LLMs were used during the idealization of the methodology or the elaboration of the results

## B    EXECUTION DETAILS

Appendix B expands the execution details to ensure reproducibility.

The dataset and all relevant code is available at https://anonymous.4open.science/r/TiEBe-3235.

## B.1 PROMPTS

### B.1.1 PROMPTS FOR QA GENERATION

We use the following prompt to generate {n_questions} question–answer pairs from each news article, avoiding questions about information that changes frequently and direct references to the article itself.

> You are an assistant responsible for creating pairs of questions and answers based on news articles. These question-answer pairs will be used to construct a dataset for evaluating knowledge from the past. Your task is to create up to {n_questions} questions and their corresponding answers based on the information in the news article. The questions should be clear and understandable, even for those who have not read the article.
> Avoid asking about information that is constantly changing or lacks a definitive answer, such as the current death toll of an event or the present status of a specific situation. Focus on questions that will remain relevant in the future.
> Use the past tense in the questions. Avoid starting with "What is..." or referring to ongoing events or situations. Refrain from asking about the current status of a particular subject, such as an agreement or situation that may change over time.
> Additionally, avoid overly specific questions. Instead, focus on broader and more meaningful information about significant events. Keep in mind that the reader will not have access to the article itself, so do not reference the article directly (e.g. "according to the article"). Emphasize the key information the article provides, and specify the point in time when an event occurred, if necessary.
> Write the questions and answers in English, regardless of the language of the article.
> Follow this format:
> Question: {question}
> Answer: {answer}
>
> Event: {event}
> Date: {date}
> News title: {title}
> News content: {content}

### B.1.2 PROMPTS FOR MODEL GENERATION

We apply the following prompt to every candidate model: each question is posed zero-shot, and the model must return an answer in a prescribed format, in the same language as the question.

> Answer the following question:
> "{question}"
> If necessary, consider the context of {region}, Provide your response in the following format:
> "Answer: your answer"

### B.1.3 PROMPTS FOR MODEL EVALUATION

We presented the following prompt to the LLM judge: we provided the question, the gold answer, and the candidate answer, and asked the judge to produce a brief 'Reasoning:' followed by 'Correct: yes — no', marking contradictions or refusals as incorrect.

> I will provide a question, an expected answer, and the candidate's answer. Your task is to verify if the candidate's answer is correct. The expected answer is the ground truth, so if the candidate's answer contradicts the expected answer or refuses to answer, it is incorrect.
> Question: "{question}"
> Expected answer: "{expected_answer}"
> Candidate answer: "{model_answer}"
> Answer in the format
> Reasoning: (your reasoning)
> Correct: (yes—no)

Table 4: Model details

| Model | Provider | Specific-version | Execution Date |
|---|---|---|---|
| GPT-4.1 | OpenAI | gpt-4.1-2025-04-14 | April 23, 2025 |
| GPT-4.1-mini | OpenAI | gpt-4.1-mini-2025-04-14 | April 23, 2025 |
| GPT-4o | OpenAI | gpt-4o-2024-08-06 | April 23, 2025 |
| Sabiá-3 | Maritaca AI | sabia-3-2024-12-11 | April 24, 2025 |
| Sabiazinho-3 | Maritaca AI | sabiazinho-3-2025-02-06 | April 24, 2025 |
| Mistral-Large | Mistral AI | Mistral-Large | April 26, 2025 |
| LLama4 Maverick | Together AI | Llama-4-Maverick-17B-128E -Instruct-FP8 | April 26, 2025 |
| Qwen 2.5 72B Instruct | Together AI | Qwen/Qwen2.5-72B-Instruct-Turbo | April 26, 2025 |
| Qwen 2 72B Instruct | Together AI | Qwen/Qwen2-72B-Instruct | April 26, 2025 |
| Deepseek V3 | Together AI | deepseek-ai/DeepSeek-V3 | April 26, 2025 |

### B.1.4 PROMPTS FOR QUESTION AND ANSWER TRANSLATION

We use the following prompt to translate questions and answers from English into the native language of each country:

> You are a professional translator. Translate the following question and answer from English to ”{language}”, the primary language spoken in ”{country}”.
> Original Question (English): ”{question}”
> Original Answer (English): ”{answer}”

### B.2 MODEL DETAILS.

As mentioned previously, we executed all our experiments through APIs. To ensure reproducibility of our results, we report in Table 4 the provider used for each model, the specific versions of each model, and the data on which the model was used.

## C DATASET STATISTICS

Appendix C compiles the main descriptive statistics of TiEBe, outlining its regional composition and overall scale. It also provides a visual overview of the benchmark's question-type distribution.

### C.1 EVENT AVAILABILITY IN WIKIPEDIA RETROSPECTIVE PAGES

Our work uses the retrospective pages of Wikipedia for each country as a starting point, and we noticed during development that such pages have a disproportional distribution. Figure 6. shows the categorization of all countries based on how many events were listed in retrospective pages in the period from 2015 to 2025.

We can see significant outliers in the US and UK, with an event count many times higher than the average. Some regions tend to have especially low event counts, such as Africa, with the majority of countries falling in the bottom two tiers of availability. Countries with too few listed events may be implausible to add to TiEBe since they would consist of too few questions. This analysis strengthens the need for more generic event sampling strategies.

### C.2 EVENTS AND EXTRACTED STATISTICS

As mentioned previously, we extracted events from Wikipedia retrospective pages of 23 different regions for 10 different years; each event in these pages may contain external references that expand

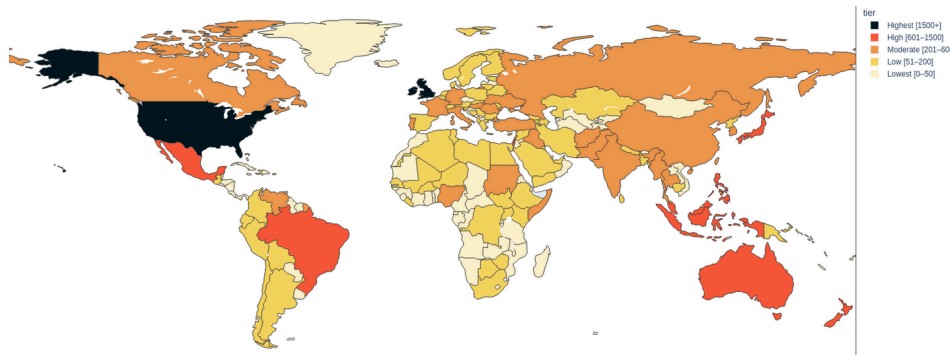

Figure 6: Categorization of countries based on the number of events available in retrospective pages between 2015 and 2025.

upon the event subject. Some events do not present such references, others present more than one. However, many of these external references no longer exist, were wrongly input in the page, or do not allow scrapers to retrieve their content, reducing the number of usable events we can use in our pipeline.

Table 5 reports the total Wikipedia events extracted, the number of references retrieved, English Q-A count and the final Q-A count per region. We retrieved 20,575 unique reference documents for 17,370 unique events. Overall, we lost around 4k events due to missing external references.

Table 5: Extraction Statistics for TiEBe.

| Region | Events | References | Extracted | QA English | QA Total |
|--------|--------|-----------|-----------|-----------|----------|
| Argentina | 225 | 190 | 140 | 135 | 270 |
| Australia | 1438 | 1955 | 908 | 811 | 811 |
| Brazil | 637 | 769 | 650 | 537 | 1074 |
| Canada | 500 | 475 | 402 | 315 | 315 |
| China | 427 | 381 | 326 | 307 | 614 |
| Colombia | 155 | 145 | 129 | 115 | 230 |
| DR Congo | 185 | 168 | 161 | 144 | 288 |
| Ethiopia | 206 | 240 | 199 | 170 | 340 |
| France | 476 | 332 | 300 | 256 | 512 |
| Germany | 374 | 293 | 269 | 223 | 446 |
| India | 852 | 784 | 699 | 550 | 1100 |
| Indonesia | 1501 | 1647 | 1552 | 1339 | 2678 |
| Mexico | 1177 | 1543 | 1077 | 982 | 1964 |
| New Zealand | 961 | 1649 | 877 | 849 | 849 |
| Nigeria | 319 | 313 | 266 | 248 | 248 |
| Papua New Guinea | 65 | 75 | 67 | 59 | 118 |
| Portugal | 255 | 279 | 258 | 228 | 456 |
| Russia | 435 | 446 | 412 | 362 | 724 |
| Turkey | 428 | 636 | 393 | 379 | 758 |
| Ukraine | 405 | 330 | 314 | 290 | 580 |
| United Kingdom | 4466 | 5443 | 4733 | 4242 | 4242 |
| United States | 3843 | 4653 | 3924 | 3228 | 3228 |
| World | 1757 | 3948 | 2519 | 1601 | 1601 |
| **Total** | **21087** | **26694** | **20575** | **17370** | **23446** |

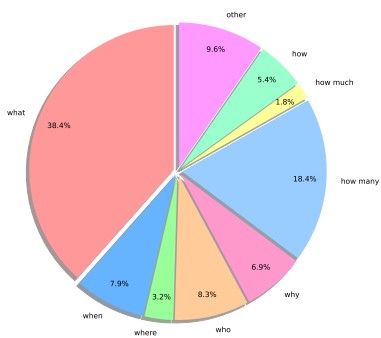

Figure 7: Distribution of question types.

### C.3 QUESTION TYPE DISTRIBUTION

We analyzed the type of questions that constitute TiEBe, Figure 7 shows the percentage of each question type; We can see a higher concentration of 'what' and 'how many' questions, with the rest reasonably well distributed.

## D TIEBE PERFORMANCE X SOCIECONOMIC INDICATORS

Appendix D presents a correlation analysis between models' average accuracy on events before 2023 and country-level socioeconomic indicators (Table 3).

These indicators–GDP (Gross Domestic Product), HDI (Human Development Index), MYS (Mean Years of Schooling), and population–are widely recognized and historically significant metrics that capture essential dimensions of national development: economic output, human development, educational attainment, and demographic scale, respectively. Their global relevance and comparability make them foundational benchmarks for cross-country analyses. For this study, we use values anchored to the beginning of our study period (2015). GDP and population data were obtained from the World Bank Open Data platform, which aggregates and standardizes economic and demographic statistics from national statistical offices and international organizations World Bank (2016; 2023). HDI and MYS data were retrieved from the United Nations Development Programme (UNDP) Human Development Reports, which compile national statistics into composite indices reflecting long-term human development trends United Nations Development Programme (2016; 2023).

Figures 8 and 9 each plot accuracy against GDP, HDI, MYS, and population in panels (a)–(d): the former for English prompts and the latter for native-language prompts.

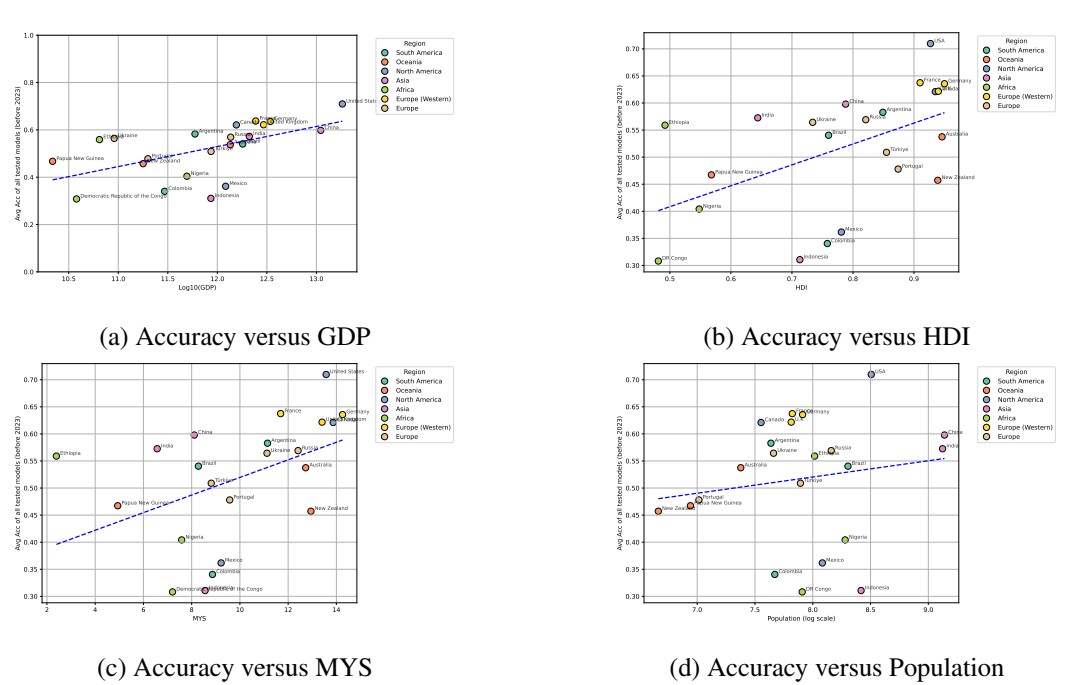

(a) Accuracy versus GDP

(b) Accuracy versus HDI

(c) Accuracy versus MYS

(d) Accuracy versus Population

Figure 8: Average accuracy of all tested models in English questions of events before 2023 versus various socieconomical indicators.

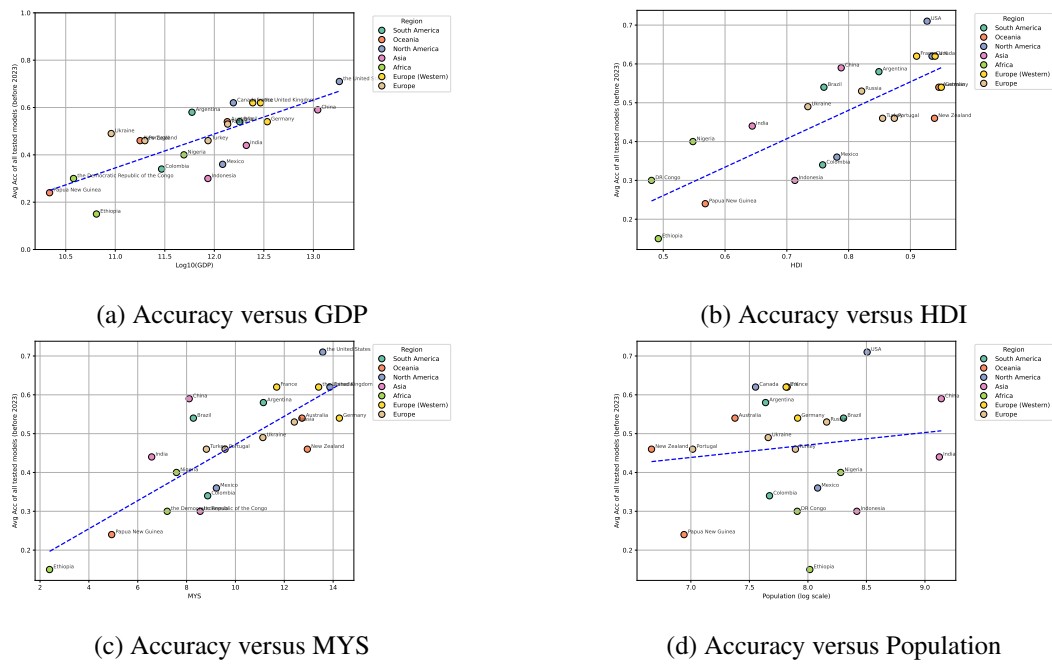

(a) Accuracy versus GDP

(b) Accuracy versus HDI

(c) Accuracy versus MYS

(d) Accuracy versus Population

Figure 9: Average accuracy of all tested models of events before 2023 in questions in their respective native languages versus various socieconomical indicators.

Table 6: Country-level socioeconomic indicators (2015).

| Country | Region | HDI | GDP (USD) | Mean Years of Schooling | Population |
|---|---|---|---|---|---|
| Argentina | South America | 0.849 | $594 749 285 413 | 11.14 | 43 477 012 |
| Australia | Oceania | 0.946 | $1 351 296 372 254 | 12.72 | 23 815 995 |
| Brazil | South America | 0.760 | $1 802 212 206 905 | 8.28 | 201 675 532 |
| Canada | North America | 0.935 | $1 556 508 816 217 | 13.86 | 35 704 498 |
| China | Asia | 0.788 | $11 061 572 618 579 | 8.11 | 1 379 860 000 |
| Colombia | South America | 0.758 | $293 492 370 193 | 8.86 | 46 969 940 |
| Ethiopia | Africa | 0.492 | $64 589 328 551 | 2.39 | 103 867 135 |
| France | Europe (Western) | 0.910 | $2 442 483 452 643 | 11.68 | 66 548 272 |
| Germany | Europe (Western) | 0.950 | $3 423 568 450 957 | 14.25 | 81 686 611 |
| India | Asia | 0.644 | $2 103 588 360 045 | 6.57 | 1 328 024 498 |
| Indonesia | Asia | 0.713 | $860 854 232 718 | 8.56 | 261 799 249 |
| Mexico | North America | 0.781 | $1 213 294 467 717 | 9.22 | 121 072 306 |
| New Zealand | Oceania | 0.939 | $178 104 220 785 | 12.94 | 4 609 400 |
| Nigeria | Africa | 0.548 | $493 026 682 801 | 7.59 | 190 671 878 |
| Papua New Guinea | Oceania | 0.568 | $21 723 437 010 | 4.93 | 8 743 246 |
| Portugal | Europe | 0.874 | $199 038 523 120 | 9.58 | 10 358 076 |
| Russia | Europe | 0.821 | $1 363 482 182 198 | 12.41 | 144 640 716 |
| Turkey | Europe | 0.855 | $864 313 810 469 | 8.81 | 78 218 479 |
| Ukraine | Europe | 0.734 | $91 030 967 789 | 11.12 | 45 784 896 |
| DR Congo | Africa | 0.481 | $37 917 706 497 | 7.21 | 81 035 531 |
| U.K. | Europe (Western) | 0.940 | $2 927 911 140 917 | 13.40 | 65 116 219 |
| USA | North America | 0.927 | $18 295 019 000 000 | 13.57 | 320 738 994 |

# E  FULL RESULTS

Appendix E collates every quantitative figure produced by our evaluation pipeline. For each country (plus the "World" category) we display a heat-map of accuracy covering the full 2015 – 2025 window. Rows correspond to the nine models under test, while columns represent calendar years. Colour intensity encodes accuracy, so reading across a row shows how a single model's recall evolves through time, whereas scanning down a column compares different models on the same year's events.

## E.1  ENGLISH QUESTIONS

In this subsection, every question is asked in English, regardless of the source article's original language. By keeping the language fixed we minimise the impact of multilingual comprehension on accuracy, so the heat-maps reflect primarily each model's factual recall. Because rows are models and columns are years, horizontal patterns reveal a model's temporal drift, whereas vertical patterns highlight which years are universally easier or harder across systems.

## E.2  NATIVE-LANGUAGE QUESTIONS

Beyond English prompts, we also test each model with questions translated into the official language of every non-English country. Using DeepSeek-V3, we translate both the QA pairs and the judge prompt, then require the model to answer in that same language. All other evaluation settings remain unchanged.

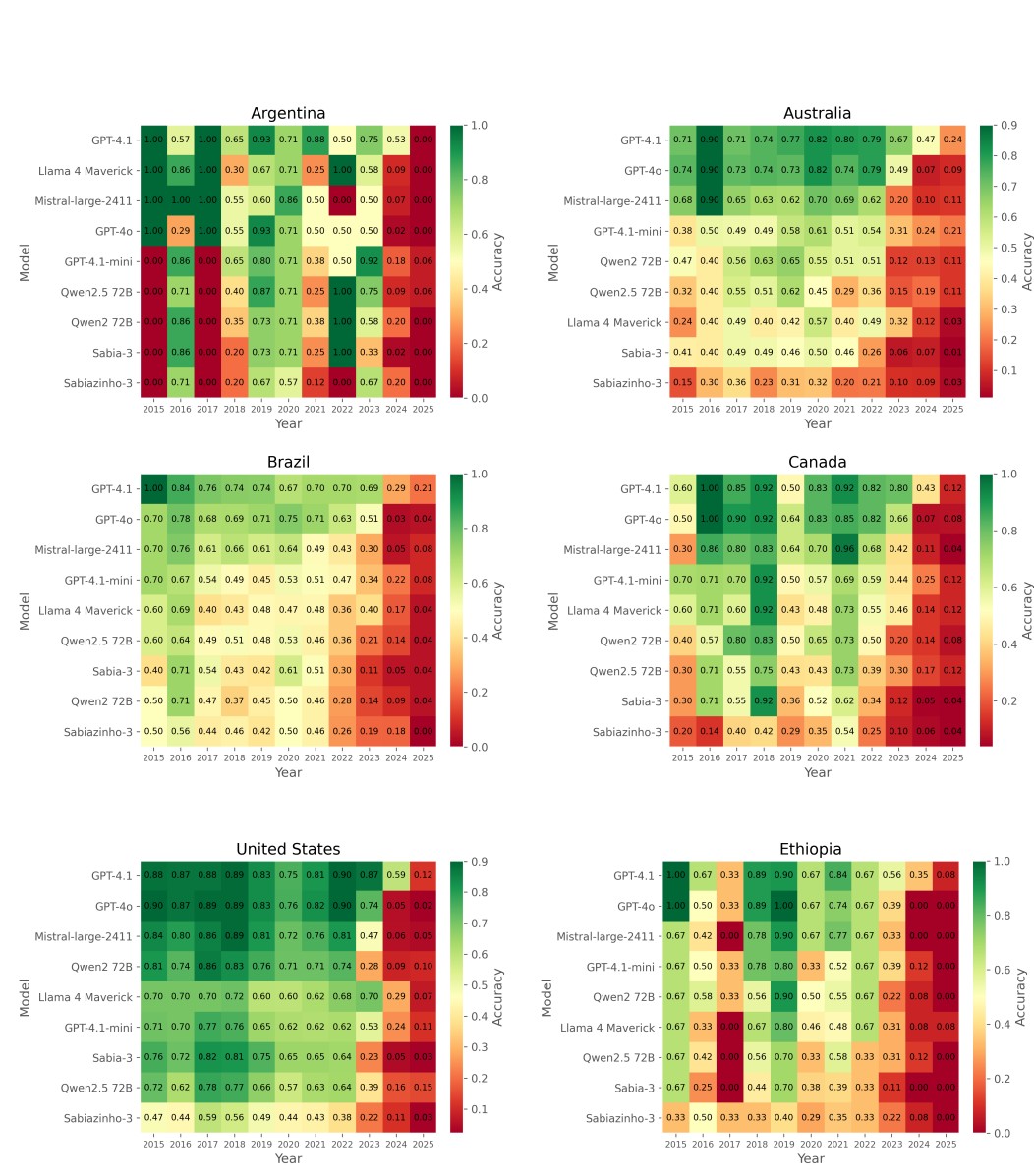

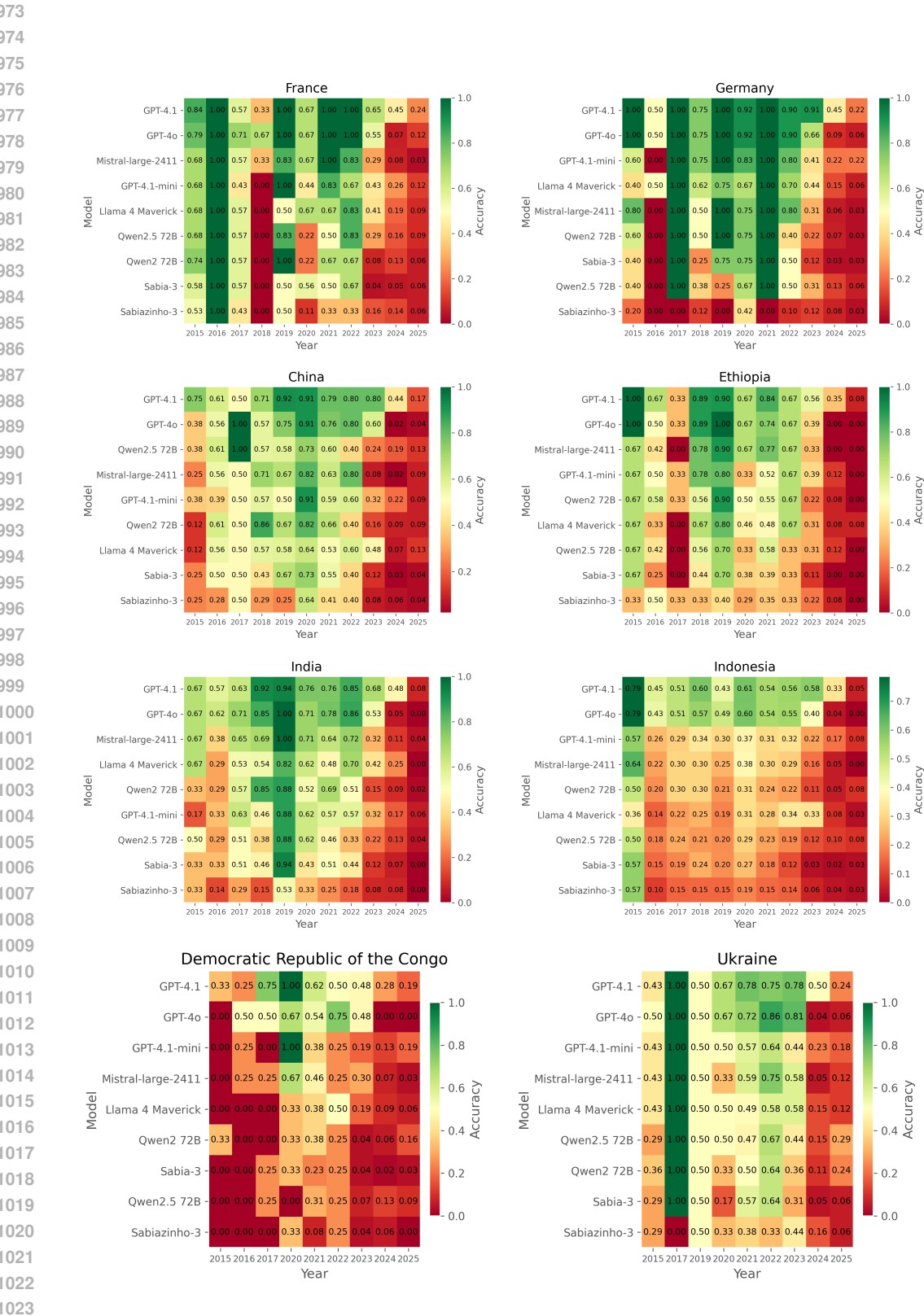

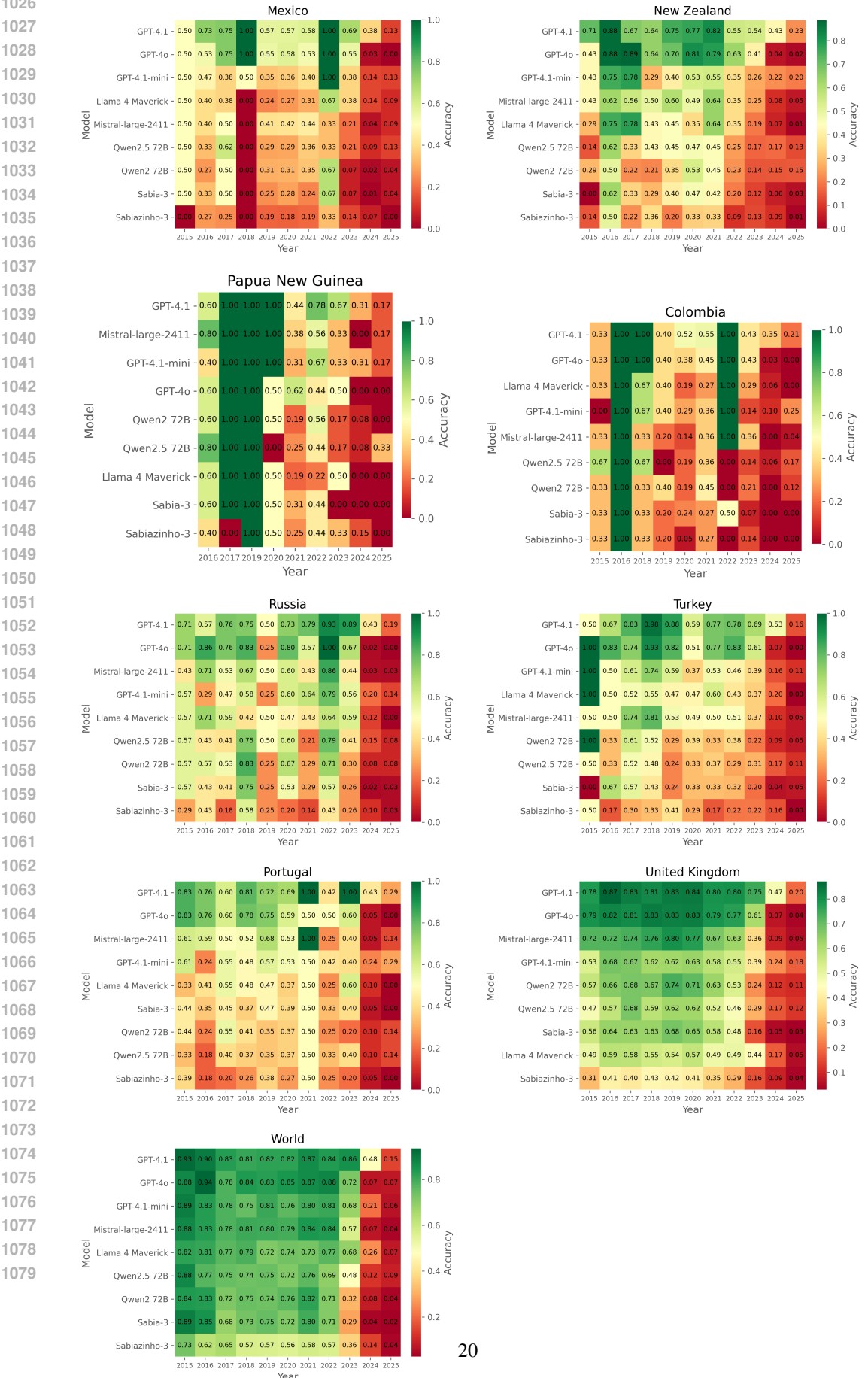

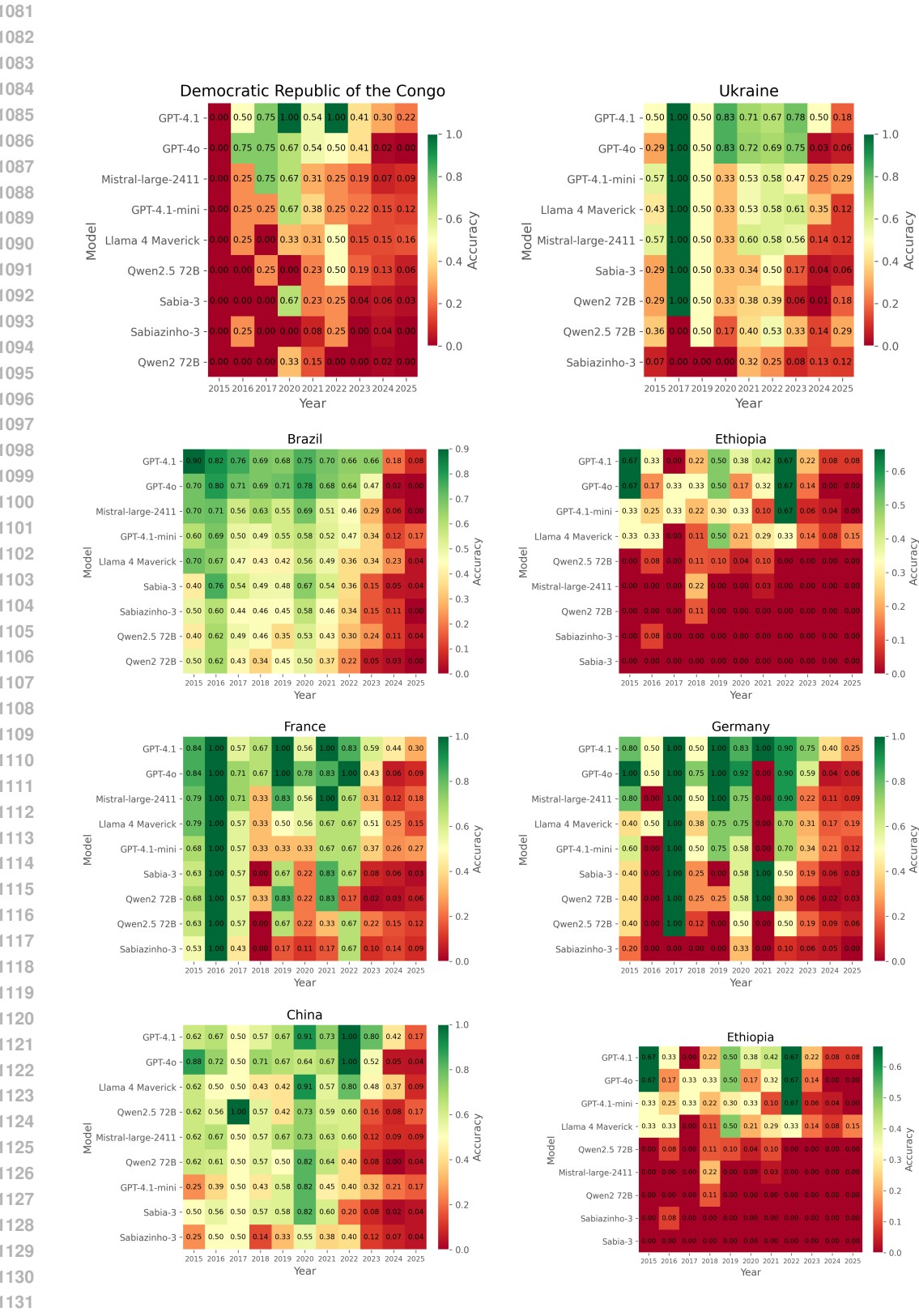

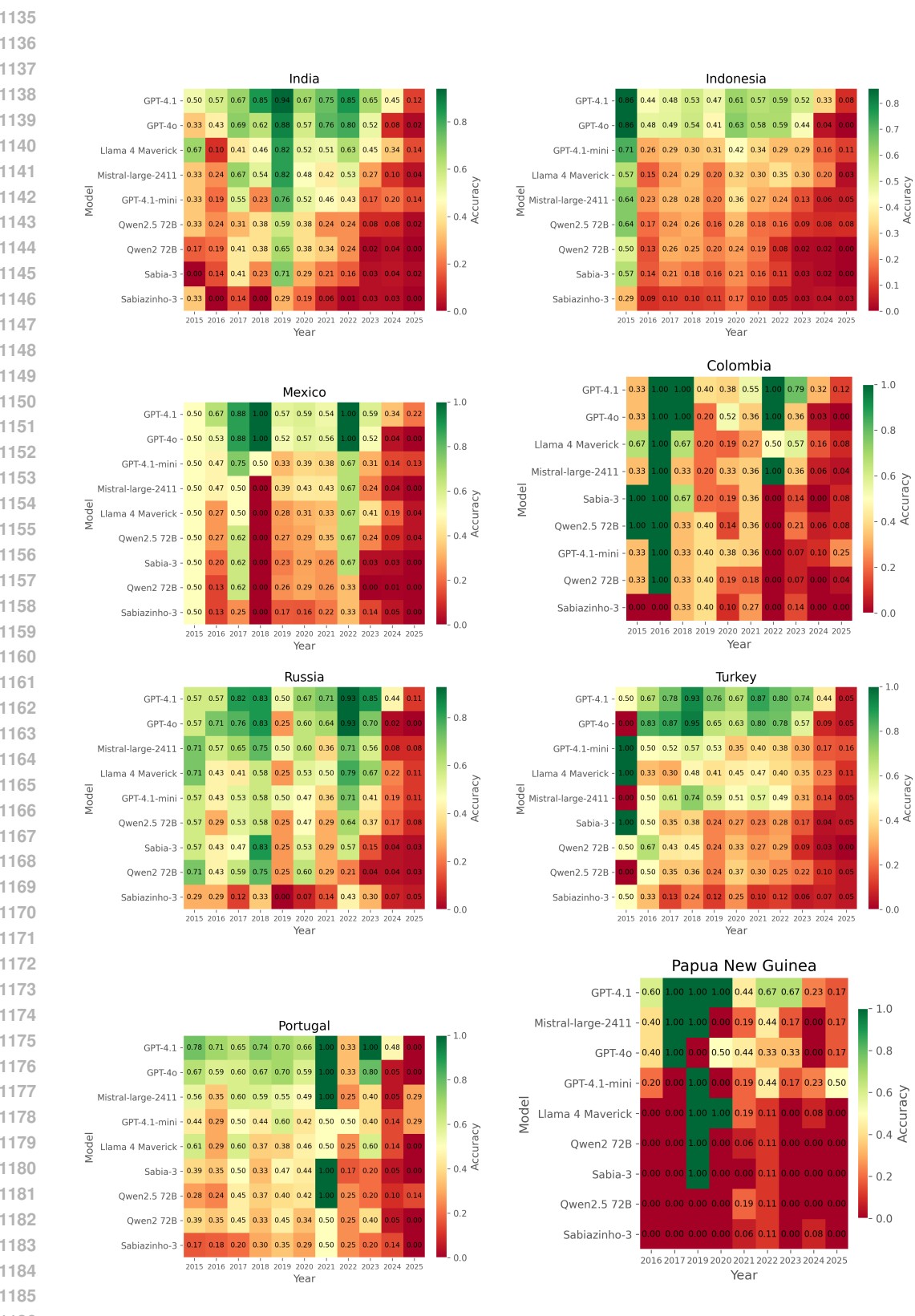

