# OpenReview forum: "TiEBe: Tracking Language Model Recall of Notable Worldwide Events Through Time"
_ICLR.cc/2026/Conference — Submitted to ICLR 2026_

### Official Review · Reviewer_dvRr · 2025-11-01

**Soundness:** 2
**Presentation:** 3
**Contribution:** 1
**Rating:** 2
**Confidence:** 4

**Summary:**

- Proposes TiEBe (Timely Events Benchmark), a large-scale multilingual and multi-regional benchmark to evaluate LLMs’ ability to recall real-world events over time.

- Covers 23 regions, 13 languages, and ~23K QA pairs spanning 2015–2025, focusing on temporal and geographic diversity.

- Provides an evaluation of various LLMs (e.g., GPT-4, Claude, Gemini, etc.), revealing strong temporal decay and regional/language disparities in factual accuracy.

**Strengths:**

- Timely and challenging idea: Capturing time-varying world knowledge in a benchmark is both important and difficult; this work makes a well-designed and meaningful attempt to address that gap.

- Broad multilingual and geographic coverage: The dataset spans many countries and includes both English and native-language questions, enabling diverse and fair evaluation across linguistic and regional contexts.

- Clear presentation: The paper is well-structured and easy to follow, with intuitive explanations of data construction and evaluation results.

**Weaknesses:**

While the main idea is valid and meaningful, the experimental depth and amount of analysis leveraging the dataset remain limited.

**W1. Lack of dataset validation**

- The paper does not clearly describe any human validation or systematic quality control for the automatically generated QA pairs, making it difficult to assess dataset reliability. Possible issues such as overlap between questions or hallucinated answers may exist.
- Similarly, the translation process across multiple languages requires validation to ensure consistency and accuracy.
- A deeper analysis of *what kinds* of time-varying information appear in the dataset and how those reflect real-world dynamics would help clarify what the benchmark actually measures.

**W2. Concerns on evaluation setting**

- The current evaluation does not appear to provide the *time period or contextual timestamp* of each question when prompting the model. However, many factual questions (e.g., *“Who is the president of [country]?”*) have time-dependent answers that can change year by year.
- Without explicitly anchoring the time context, model errors may reflect ambiguity in the question rather than a genuine knowledge gap.

**W3. Limited evaluation depth**

- Although the benchmark focuses on *time-dependent knowledge*, the analysis does not fully exploit this aspect. Temporal variance is observed but not deeply explored (e.g., how recall changes across time windows or event types).
- As mentioned in W2, certain facts can change over time (e.g., political positions or economic indicators). Analyzing such cases could provide valuable insight into how models handle *conflicting or updated knowledge* during training, i.e., whether new information overwrites or coexists with older facts.
- Moreover, the results in Section 4.2 are not particularly surprising, likely because the evaluated models (mostly released in 2023–2024) have limited exposure to post-2023 factual data.
- Instead of only reporting numerical scores, the paper could highlight *what we can learn* from these temporal gaps, for example, how LLMs adapt (or fail to adapt) to evolving world knowledge and where the main bottlenecks lie.

**W4. Insufficient explanation in Section 4.3**

- The discussion on performance drops in *Tok Pisin* and *Amharic* seems inconsistent with the reported results: models appear to perform worse even when queried in English.
- A few qualitative examples or error analyses are needed to clarify and support the authors’ interpretation.

**Questions:**

Revealed in the Weaknesses part

---

> ### Author Response · Authors · 2025-12-03
>
> First, we would like to thank the reviewer for the comments.
>
> Regarding the mentioned weakness:
>
>  We would like to clarify that time-dependent questions explicitly include the specific year. Figure 2 of the paper presents examples of such questions. Additionally, as an extra safeguard, Appendix B.1.2 shows the prompt used to generate answers and provides further context about the region being discussed.
>
> The reviewer writes: “Although the benchmark focuses on time-dependent knowledge, the analysis does not fully exploit this aspect. Temporal variance is observed but not deeply explored (e.g., how recall changes across time windows or event types).”
>  We believe that Figure 4 demonstrates how performance shifts across different time windows. We agree, however, that analyzing accuracy across different types of questions could be an interesting extension.
>
> The reviewer also writes: “Instead of only reporting numerical scores, the paper could highlight what we can learn from these temporal gaps, for example, how LLMs adapt (or fail to adapt).”
>
>  We do not fully understand this comment. In the paper, we show that the primary gap is not temporal but regional. The reviewer further states: “The results in Section 4.2 are not particularly surprising, likely because the evaluated models (mostly released in 2023–2024) have limited exposure to post-2023 factual data.” While this is true, we believe this is actually evidence that our methodology works as expected, since models cannot reasonably recall events they were not exposed to during pretraining.
> Finally, regarding the claim: “The discussion on performance drops in Tok Pisin and Amharic seems inconsistent with the reported results: models appear to perform worse even when queried in English.”
>  It is correct that models perform worse for these regions even when the questions are asked in English. However, it is also true that these gaps become even larger when the questions are presented in the original languages.

---

### Official Review · Reviewer_HVLJ · 2025-11-03

**Soundness:** 2
**Presentation:** 1
**Contribution:** 2
**Rating:** 2
**Confidence:** 3

**Summary:**

This paper works on the challenge of evaluating how LLMs recall factual knowledge about world events across time and geographic regions. The authors introduce the Timely Events Benchmark (TiEBE), a dataset with over 23,000 question-answer pairs spanning ten years, 23 regions, and 13 languages.  Key experimental results from evaluating nine different LLMs show significant geographic disparities in factual recall. Performance is strongly correlated with socioeconomic indicators like a country's Gross Domestic Product (GDP) and Human Development Index (HDI), with Spearman correlations exceeding 0.7.

**Strengths:**

1.  The paper presents a scalable method for benchmark creation by using Wikipedia retrospective pages to identify notable events, ensuring a structured and continuously updatable data source for temporal analysis.
2.  Experimental evaluation is extensive, testing nine different open-source and commercial LLMs. This provides a broad and representative assessment of current model capabilities and their shared limitations.

**Weaknesses:**

### About Method

1. The paper only validates LLM-as-judge reliability but lacks human evaluation of the generated QA pairs themselves. LLM-generated questions may contain factual errors or inconsistencies. Recommend conducting human evaluation on a sample to assess quality metrics like factual consistency and answerability.
2.  GPT-4o achieved 91% consistency versus DeepSeek-V3's 88.5%, yet the paper chose the lower-performing DeepSeek-V3 as judge without explanation (cost, API availability, etc.). Should clarify the rationale or use the better-performing model.
3. The limitation section mentions contamination risks but lacks depth. Since the dataset uses Wikipedia content, models likely encountered it during pre-training, potentially testing memorization rather than reasoning. Need deeper analysis and mitigation strategies.

### About Experiment

1.  The evaluation relies heavily on DeepSeek-V3 as a judge, yet the authors' own analysis indicates that GPT-4o has higher agreement with humans. The authors should justify the choice of the slightly inferior judge, and include a sensitivity analysis using GPT-4o as the judge to verify the robustness of the core conclusions to this choice.
2.  For a new benchmark paper, the experimental section critically overlooks the issue of data contamination. The authors should conduct a preliminary contamination analysis, for instance, by searching for TiEBE test samples in the training data of open-source models, to quantify potential train-test overlap; otherwise, it is difficult to discern whether models are performing factual recall or simple pattern matching.
3.  The paper lacks a qualitative analysis of the models' specific failure cases. It is recommended to add a section that presents and analyzes typical incorrect answers from models across different regions, languages, or time periods, which would provide deeper insights into the models' capability boundaries than accuracy scores alone.

**Questions:**

None

---

> ### Author Response · Authors · 2025-12-03
>
> First, we would like to thank the reviewer for the comments.
>
> Regarding the mentioned weakness:
>  We chose to exclude a model from the evaluation pool (in this case, DeepSeek-V3) to avoid concerns that a model judging itself could introduce bias toward assigning higher scores. For example, if GPT-4o were the judge, doubts could be raised about the impartiality of its own evaluation. For this reason, we opted to use DeepSeek-V3 as the judging model. A second consideration was cost: using DeepSeek-V3 for evaluation was substantially cheaper, which allowed us to test a larger number of models. It is worth noting that TiEBe contains 17k QA pairs; evaluating 10 models therefore requires approximately 170k requests to the judge model.
> The reviewer writes: “The limitation section mentions contamination risks but lacks depth. Since the dataset uses Wikipedia content, models likely encountered it during pre-training, potentially testing memorization rather than reasoning. Need deeper analysis and mitigation strategies.”
>
>  We believe this interpretation misses the purpose of TiEBe. Our goal is to track notable events, and we fully expect such events to have appeared in the models’ pretraining data. The benchmark is designed to study the gap in exposure and recall of these events across different regions of the world. Thus, contamination is not a primary concern: although the underlying events are likely present in pretraining corpora, the questions themselves are entirely new, and what we aim to measure is how well models recall events that they were presumably already exposed to—and how this varies geographically.

---

### Official Review · Reviewer_CCW7 · 2025-11-05

**Soundness:** 1
**Presentation:** 2
**Contribution:** 1
**Rating:** 2
**Confidence:** 4

**Summary:**

This paper introduces TiEBe (Timely Events Benchmark), a new dataset with over 23,000 question-answer pairs designed to evaluate an LLM's factual recall of notable world events across time, geography, and language. Covering 10 years and 23 global areas, the benchmark exposes substantial geographic inequalities in model performance, with factual accuracy showing a strong positive correlation (Spearman > 0.7) with socioeconomic metrics. The results further reveal that although models achieve reasonable accuracy in many languages, their performance deteriorates significantly on low-resource languages like Tok Pisin and Amharic, highlighting persistent systemic biases in current LLMs.

**Strengths:**

The paper is grounded in strong motivation and tackles an important and relevant problem. Building a benchmark to evaluate time-sensitive world knowledge in LLMs fills a major gap in current evaluation practices, making this contribution meaningful and well-justified.

**Weaknesses:**

1. Unclear justification for using only DeepSeek-V3: The rationale behind selecting DeepSeek-V3 as the sole model for generating questions, translations, and evaluations is not well supported. Although the authors claim the model performs adequately, their own results (Table 2) show that GPT-4o aligns more closely with human judgments. Relying exclusively on one model—especially a less optimal one—seems unjustified. A more credible design would validate outputs across multiple models or through an ensemble approach.
2. No secondary validation of generated data: The paper does not describe any additional quality control step to verify the QA pairs generated by DeepSeek-V3. Without screening for hallucinations, factual errors, or model-induced biases, the reliability and correctness of the dataset remain questionable.
3. Limited novelty in the analysis: Although the dataset itself is useful, the analytical results offer little novelty beyond what is already obvious.
4. Conclusions are obvious and potentially problematic: The main findings—models perform worse on low-resource languages and better for countries with stronger AI ecosystems—are expected. Moreover, drawing direct correlations with socioeconomic metrics like GDP or HDI is oversimplified. It ignores the primary factor: disparity in available training data. This kind of interpretation risks reinforcing harmful biases or socioeconomic stereotypes.
5. Overly binary evaluation method: The authors mention that the LLM judge is often stricter than human annotators, yet they still rely on a binary Correct/Incorrect scoring scheme. This is inadequate. A minor temporal mismatch (e.g., a correct answer from a different year) is not comparable to a completely wrong answer or refusal to answer. A more fine-grained evaluation scale is needed to reflect these distinctions.

**Questions:**

See weaknesses

---

> ### Author Response · Authors · 2025-12-03
>
> First, we would like to thank the reviewer for the comments.
> Regarding the mentioned weakness:
>  We chose to exclude a model from the evaluation pool (in this case, DeepSeek-V3) to avoid concerns that a model judging itself could introduce bias toward assigning higher scores. For example, if GPT-4o were the judge, questions could be raised about the impartiality of its evaluations. We agree that evaluating using multiple models would be interesting; however, this comment seems to overlook the cost of doing so. TiEBe is composed of more than 17k questions, and repeating the evaluation for every model across all questions would be financially unfeasible for most research groups.
> The reviewer writes: “The main findings—models perform worse on low-resource languages and better for countries with stronger AI ecosystems—are expected. Moreover, drawing direct correlations with socioeconomic metrics like GDP or HDI is oversimplified. It ignores the primary factor: disparity in available training data.”
>  We feel this characterization somewhat misrepresents our effort. On average, models recalled 12% fewer events from Portugal than from Canada. Is that truly expected? We do not believe so. Furthermore, the notion of “countries with stronger AI ecosystems” is subjective and not clearly defined. Additionally, most state-of-the-art models do not currently disclose their training sets, and we believe that quantifying the presence of each language in the pretraining ecosystem would, on its own, constitute an entire research paper.

---

### Official Review · Reviewer_mRZh · 2025-11-07

**Soundness:** 2
**Presentation:** 2
**Contribution:** 2
**Rating:** 2
**Confidence:** 4

**Summary:**

This work presents a dataset of automatically generated questions about notable global and regional event that are generated from wikipedia articles. The authors prompt the model to generate questions from different countries wikipedia entry snapshots taken from different time periods, spanning a decade. The authors then evaluate LMs on their constructed dataset, demonstrating trends in models being able to answer questions better for events from some countries more than others (e.g., models answer questions about US events better than those about Indonesia) and models tend to struggle with facts from most recent years.

**Strengths:**

This work presents a large dataset of questions about events from different countries and times. One distinction from prior works establishing similar datasets is the inclusion of questions in different languages reflecting the country the question pertains to.

**Weaknesses:**

1. Limited discussion of related work, both in the related work section and in the paper as a whole. In particular, lost of prior work has explored the impact of temporal or geographical context on wikipedia-based factoid QA [1]. Other works have looked at both individually [2, 3]. Additionally other works have developed other methods of synthetically identifying such temporally dependent QA pairs or facts from Wikipedia pages or other related resources like wikidata [4, 5]. Other works have looked into the impact of socioeconomic factors on LM behavior or performance. Given the overlap in task, methods, and findings, such works should be discussed, compared against, or evaluated on.

[1] SituatedQA: Incorporating Extra-Linguistic Contexts into QA
Michael J.Q. Zhang, and Eunsol Choi.
EMNLP 2021

[2] Visually Grounded Reasoning across Languages and Cultures
Fangyu Liu, Emanuele Bugliarello, Edoardo Maria Ponti, Siva Reddy, Nigel Collier, Desmond Elliott
EMNLP 2021

[3] Time Waits for No One! Analysis and Challenges of Temporal Misalignment
Kelvin Luu, Daniel Khashabi, Suchin Gururangan, Karishma Mandyam, Noah A. Smith
NAACL 2022

[4] A Dataset for Answering Time-Sensitive Questions
Wenhu Chen, Xinyi Wang, William Yang Wang
NEURIPS 2021

[5] Time-Aware Language Models as Temporal Knowledge Bases
Bhuwan Dhingra, Jeremy R. Cole, Julian Martin Eisenschlos, Daniel Gillick, Jacob Eisenstein, William W. Cohen
TACL 2021

[6] Whose Opinions Do Language Models Reflect?
Shibani Santurkar, Esin Durmus, Faisal Ladhak, Cinoo Lee, Percy Liang, Tatsunori Hashimoto


2. The constructed QA pairs

**Questions:**

Are all the all the results on non-English languages from models that have been trained on the target languages? Or have the developers claimed that model should support the languages?

---

> ### Author Response · Authors · 2025-12-03
>
> First, we would like to thank the reviewer for the comments.
> Regarding the mentioned weaknesses:
>  We agree that some of the cited papers can be incorporated into the related work and discussed. However, we believe that most of the works referenced by the reviewer do not diminish TiEBe’s contribution. Our benchmark focuses on how recent LLMs recall notable events, rather than general Wikipedia QA information, and analyzes this phenomenon both geographically and temporally. We do not see any work among those cited by the reviewer that conducts a similar study.
> The reviewer mentions “constructed QA pairs” as a weakness, but no clarification or details are provided. We believe this may have been a typo or misunderstanding. Without additional context, we are unable to properly address this point.
> Regarding the reviewer’s questions:
>  Most state-of-the-art models today do not clearly specify the languages they support, often claiming to be simply “multilingual”. Regardless, we believe it is important to study in more detail the capabilities of notable LLMs in different languages and regions, in order to better inform their appropriate usage.

---

### Meta-Review · Area_Chair_pXsj · 2025-12-24

**Summary:**

The paper introduces TiEBe (Timely Events Benchmark), a dataset of over 23,000 question-answer pairs designed to evaluate Large Language Models' (LLMs) factual recall of notable world events spanning ten years, 23 regions, and 13 languages. The authors utilize Wikipedia retrospective pages to construct these QA pairs and evaluate nine models, revealing significant geographic disparities in performance that correlate strongly with socioeconomic indicators like GDP and HDI.

The reviewers generally appreciated the motivation and the scale of the benchmark, acknowledging the difficulty of capturing time-varying world knowledge. However, there was a consensus of concern regarding the experimental rigor, specifically the reliance on a single, less capable model (DeepSeek-V3) for the entire pipeline (generation, translation, and judging) without human validation of the generated dataset.

Based on the reviewer feedback, the AC recommends Reject, primarily due to the lack of human validation for the synthetically generated dataset content, which undermines the reliability of the benchmark despite its valuable motivation and scale.

**Reviewer Concerns:**

**Addressed Concerns:**
- Data Contamination (Reviewer HVLJ): The reviewer argued that using Wikipedia data tests memorization rather than reasoning. The rebuttal successfully clarified that the benchmark’s specific goal is to test recall of notable events exposed during pre-training, making this critique a misunderstanding of the paper's objective.
- Missing Timestamps (Reviewer dvRr): The reviewer believed prompts lacked specific time contexts (e.g., "Who is the president?"). The authors effectively pointed to Figure 2 and Appendix B.1.2 to demonstrate that questions are explicitly anchored to specific years.
- Judge Model Selection (Reviewers CCW7, HVLJ): Reviewers criticized using DeepSeek-V3 over the more accurate GPT-4o. The authors provided a reasonable defense regarding the prohibitive cost of using GPT-4o for a dataset of this size and the methodological soundness of avoiding self-evaluation bias.

**Outstanding Concerns:**
- Dataset Validation (Reviewers CCW7, HVLJ, dvRr): This remains the most significant unaddressed weakness. While the authors defended the judge, they did not address the lack of human validation for the generated QA pairs themselves. There is no guarantee that the synthetic questions are free of hallucinations or factual errors, which is critical for a benchmark paper.
- Binary Evaluation Metrics (Reviewer CCW7): The concern that a strict Correct/Incorrect metric fails to capture nuances (such as minor temporal mismatches) was not addressed in the rebuttal.
- Qualitative Analysis (Reviewer HVLJ): The request for a deeper analysis of failure cases to provide insight beyond raw accuracy numbers remains unfulfilled.

**Reviewer Scores:**

- Reviewer mRZh: (2 to 3). The author agreed to include the missing related work.
- Reviewer CCW7: (2 to 2). The authors provided a strong practical defense for the judge model choice regarding cost and bias. However, the critical lack of validation for the synthetic dataset content remains a blocker for a higher score.
- Reviewer HVLJ: (2 to 3). This reviewer's primary negative critique regarding "contamination" was effectively refuted as a misunderstanding of the task (recall vs. reasoning). With that resolved, the evaluation of the paper's contribution should improve slightly.
- Reviewer dvRr: (2 to 3). The rebuttal addresses the concern that prompts lacked timestamps by pointing to specific figures in the text. Correcting this oversight should lead to a more favorable assessment of the methodology. However, the lack of validation concern remains.

---

### Decision · Program_Chairs · 2026-01-26

Reject